# MetaboListem and TABoLiSTM: Two Deep Learning Algorithms for Metabolite Named Entity Recognition

**DOI:** 10.3390/metabo12040276

**Published:** 2022-03-22

**Authors:** Cheng S. Yeung, Tim Beck, Joram M. Posma

**Affiliations:** 1Section of Bioinformatics, Division of Systems Medicine, Department of Metabolism, Digestion and Reproduction, Faculty of Medicine, Imperial College London, London SW7 2AZ, UK; c.yeung20@imperial.ac.uk; 2Department of Genetics and Genome Biology, University of Leicester, Leicester LE1 7RH, UK; 3Health Data Research (HDR), London NW1 2BE, UK

**Keywords:** deep learning, named entity recognition, natural language processing

## Abstract

Reviewing the metabolomics literature is becoming increasingly difficult because of the rapid expansion of relevant journal literature. Text-mining technologies are therefore needed to facilitate more efficient literature reviews. Here we contribute a standardised corpus of full-text publications from metabolomics studies and describe the development of two metabolite named entity recognition (NER) methods. These methods are based on Bidirectional Long Short-Term Memory (BiLSTM) networks and each incorporate different transfer learning techniques (for tokenisation and word embedding). Our first model (MetaboListem) follows prior methodology using GloVe word embeddings. Our second model exploits BERT and BioBERT for embedding and is named TABoLiSTM (Transformer-Affixed BiLSTM). The methods are trained on a novel corpus annotated using rule-based methods, and evaluated on manually annotated metabolomics articles. MetaboListem (F1-score 0.890, precision 0.892, recall 0.888) and TABoLiSTM (BioBERT version: F1-score 0.909, precision 0.926, recall 0.893) have achieved state-of-the-art performance on metabolite NER. A training corpus with full-text sentences from >1000 full-text Open Access metabolomics publications with 105,335 annotated metabolites was created, as well as a manually annotated test corpus (19,138 annotations). This work demonstrates that deep learning algorithms are capable of identifying metabolite names accurately and efficiently in text. The proposed corpus and NER algorithms can be used for metabolomics text-mining tasks such as information retrieval, document classification and literature-based discovery and are available from the omicsNLP GitHub repository.

## 1. Introduction

Since the late 1990s, metabolomics has contributed to better the understanding of the roles that small molecules play in health and disease [1]. The on-going developments of high-throughput data acquisition ensures that the technology is accessible to more researchers [2], evident by the increasing number of studies year-on-year for studies in any organism as well as those only done in humans (Appendix A). With the growth of the number of publications comes the difficulty of literature review, and performing a literature review of all papers relating to a specific phenotype has become an almost impossibly difficult task for researchers [3]. To facilitate more efficient review processes, computational literature mining tools are needed.

Natural language processing (NLP) is a branch of computer science that aims to understand natural human language computationally; in the biomedical field, NLP often refers to the technologies that allow computational tools to interpret scientific texts written by humans. In recent years, NLP has been successfully applied in many biomedical text mining tasks, including information retrieval, document classification and literature-based discovery [4,5,6,7]. By identifying biologically important entities in articles and computationally inferring the connection between them, NLP facilitates understanding of biological relationships from textual information. Such efficient and automatic knowledge mining systems can provide a comprehensive viewpoint to researchers by analysing a vast amount of articles.

Some advances in metabolomics have been made for using computer algorithms to read scientific text [8,9,10,11], however these suffer from a few drawbacks such as using rule-based annotation methods and dictionary matching and corpora and code/models being unavailable. While these methods can indeed correctly capture a vast amount of metabolite names, they are dependent on the library of compound names used—and no library is exhaustive. The latest version of the Human Metabolome Database (HMDB) [12] reports over 18,557 metabolites have been detected and quantified in the human body, and 91,822 more are expected or predicted. Moreover, these numbers are still increasing, from 2180 (HMDB 1.0 in 2007) to 114,100 (HMDB 4.0 in 2017) in a decade [13] and predicted to be an order of magnitude higher [1]. Hence there is a need for generalisable approaches for automated metabolite recognition.

Within NLP, Named Entity Recognition (NER) is a field concerned with automatically detecting specific entities from texts. To facilitate biomedical text-mining, and NER in particular, BioCreative (Critical Assessment of Information Extraction in Biology) [14] have proposed multiple tasks/challenges and provided corpora for chemical entity recognition, such as the CHEMDNER (chemical compound and drug name recognition, BioCreative V4) [15] and CEMP (Chemical Entity Mention in Patents, part of BioCreative V5) [16] challenges. The use of machine learning techniques such as Conditional Random Fields (CRF) has shown promising results with F1-scores > 0.87 of tmChem [17] in the V4 challenge, however the CRF methods were surpassed by a Bidirectional Long Short-Term Memory (BiLSTM) recurrent neural networks CollaboNet [18] (F1-score of 0.88) and LSTMVoter [19] (F1-score of 0.90), and an attention-based BiLSTM [20] (Att-ChemdNER, F1-score of 0.91). For the V5 challenge, and the CEMP task, LSTMVoter achieved an F1-score of 0.89 and another deep learning (DL) system using BiLSTMs called ChemListem [21] achieved an F1-score of 0.90.

Adding a word embedding layer to NER DL models is well known to improve performance [22], and ChemListem used context-free word embeddings/representation using global vectors (GloVe) [23] for this purpose. However, in recent years big improvements have been made in NLP by using contextual embedding with transformers such as BERT (Bidirectional Encoder Representations from Transformers) [24]. BERT was trained on a corpus containing 3.3 billion words mainly from the English Wikipedia and BooksCorpus. Unlike the context-free embedding techniques, BERT’s contextual embedding makes the vector representation of a word dependent on its context. Therefore, for biomedical text mining the BioBERT model [25] was pretrained on PubMed abstracts and PubMed Central (PMC) articles [26], and has been used to obtain F1-scores on the BioCreative V4 and V5 of 0.93 and 0.94, respectively, and F1-scores of 0.78–0.85 for recognising genes and proteins [27]. However, fine-tuned BERT-based models do not always yield state-of-the-art results as they greatly depend on the training data size [28].

There are three main issues with using these chemical NER methods for metabolite NER. First, as mentioned, with these methods the context matters and therefore training algorithms to recognise chemical entities in abstracts does not necessarily mean they work as well for full-text articles. Second, while a metabolite is a chemical entity, a chemical entity is not necessarily a metabolite, therefore these existing corpora and algorithms may result in many false positives when using them in metabolomics. Last, each metabolite can have multiple (sometimes ten or even a hundred) different synonyms—over 1.2 million synonyms are reported in HMDB)—adding to the complexity of using a dictionary of synonyms as new names can be used by different authors.

Our contribution is the development of a standardised, machine-readable metabolomics corpus of full-text Open Access (OA) PMC articles by using the Auto-CORPus (Automated and Consistent Outputs from Research Publications) package [29] for text standardisation in conjunction with semi-automatic annotation of metabolites in full texts using a combination of dictionary searching (using HMDB to stay in line with prior work [10]), regular expression matching and rule-based approaches. We then use this corpus to train three DL-based algorithms (with different word embedding layers) to perform metabolite NER with the aim to obtain a generalisable model that can be used to speed up metabolomics literature review. Our work addresses these issues by first creating a publicly available corpus of full-text articles with annotated metabolites and, second, developing new DL algorithms trained to focus specifically on recognising metabolites as well as not constraining these algorithm to abstracts (like prior work), but on full-text paragraphs.

## 2. Results

Here we describe the characteristics of the automatically annotated metabolomics corpus. This corpus was then used to train three NER algorithms and the manually annotated test set used to evaluate the accuracy of the annotation pipeline, a prior algorithm for chemical NER and the three NER algorithms. Last, we applied the best performing model on all cancer metabolomics articles in the corpus to demonstrate how the algorithms can be used for literature review.

### 2.1. Metabolomics Corpus

The metabolomics corpus generated by means of the annotation pipeline (see Section 4) is of similar size to that of the BioCreative V5 dataset. We extracted 57,684 sentences that contain metabolites from 1217 metabolomics articles, and in these have identified a total of 123,658 metabolites. For comparison, the BioCreative V5 dataset contains 21,000 abstracts and 99,632 chemical entities. Table 1 and Table 2 detail the metabolomics corpus and the number of annotated unique entities for journals and traits, respectively, and illustrates that most metabolomics articles report between 20 and 30 unique metabolites. While metabolomics articles on cancer are the largest single group within the corpus, the average number of metabolites reported in cancer articles is similar to that of other disease areas. The workflow with the structure of the metabolomics corpus is shown for an abstract of a publication in Figure 1.

### 2.2. Model Performance

The metabolomics corpus was split into a training set (1034 articles, 49,007 sentences, 105,335 annotations) and a test set (183 articles). The test set was manually annotated and consists of 8950 sentences with 19,138 annotations. We first evaluated the application of the rule-based annotation pipeline on the independent test set articles. Our pipeline yields an F1-score of 0.8893, with precision and recall of 0.8850 and 0.8937 respectively (see Table 3). Next, we evaluated the performance of ChemListem [21], trained on the BioCreative V5 dataset, on the metabolomics corpus test set where it achieves an F1-score of 0.7669. Our first DL model uses the same architecture as ChemListem but was trained on the metabolomics training corpus, therefore this model was named MetaboListem, and it achieved an F1-score of 0.8900 (precision of 0.8923 and recall of 0.8877). Our second DL model uses a different embedding layer (BERT or BioBERT) to MetaboListem, but was trained on the same training dataset. The TABoLiSTM (Transformed-Affixed Bidirectional LSTM) model with BERT embeddings achieved an F1-score of 0.9004, with precision and recall rates of 0.9187 and 0.8829 respectively; the same model with BioBERT embeddings results in a 0.85% improvement of the F1-score.

The workflow including the structure of the metabolomics corpus, the annotation and result of the TABoLiSTM application on the abstract of a publication is shown in Figure 1. We exemplify the differences in the output from the TABoLiSTM model and our annotation pipeline using three snippets of text in Figure 2 that illustrate the typical examples of how the DL model predictions differ from the rule-based annotations, including the abstract from Figure 1 shown in Figure 2A. These examples include spelling errors, metabolites not included in the dictionary and metabolite abbreviations.

Beyond prediction accuracy, another aspect to evaluate is the model sizes. Despite the performance advantages, TABoLiSTM is much larger than both ChemListem and MetaboListem; the sizes of the latter two models are around 26 MB, whereas the size of the TABoLiSTM weight file exceeds 412 MB. Furthermore, pretrained BERT/BioBERT (cased versions) are required to employ the TABoLiSTM model, which adds an extra 415 MB.

The performance of each of the 5 models was evaluated for separate sections (abstract, materials/methods, results, results&discussion, discussion) in the test set (Figure 3). Across all models, the best performance was achieved on separate results and discussion sections, with lowest performance for material/methods and results&discussion sections.

### 2.3. Automated Literature Review of the Metabolomics Corpus

We applied TABoLiSTM on the abstract, results (including table data) and discussion sections of each of the 8 traits (see Table 2) of the metabolomics corpus to summarise the application of the algorithm in literature review. Figure 4 visualises the results for the 492 cancer articles in the corpus, where only metabolites found in at least 3 different articles are visualised. The size of the text of each metabolites is proportional to the number of articles that reported the metabolite and the position is based on how frequent metabolites are found together. In total, 664 metabolites are recognised in at least three of the 492 articles on cancer (compared to 12,662 unique metabolite names across all 492 articles). The five most frequent metabolites amongst these are glucose, glutamine, lactate, alanine and glutamate, which are mentioned in 22–29% of articles (see Table 4).

In the cancer summary graph, no apparent patterns or clusters are observable based on the node colours due to the three-article threshold applied to the graph. This is in contrast to other networks such as one created for the smoking trait where the restriction of 3 articles is not applied and all 1360 metabolites reported in 124 articles are visualised (see Appendix A). This reflects the low number of common metabolites for this trait in the corpus (Table 4). Glucose is the only metabolite that appears in the top 10 for all traits, with it being reported in 62% of articles on metabolic syndrome.

## 3. Discussion

We have described three new DL models, using different word embedding layers, that have both achieved state-of-the-art performance (F1-score > 0.89) for metabolite NER on a manually annotated dataset with OA metabolomics articles. Our DL models (F1-scores of 0.91, 0.90 and 0.89 for TABoLiSTM (BioBERT), TABoLiSTM (BERT) and MetaboListem, respectively) surpass by a considerable margin previous methods (using CRFs) for metabolite NER which achieved maximum F1-scores of 0.78 [8] and 0.87 [9], albeit these methods were trained and evaluated on different corpora. The full corpora for both of these methods are not available, only the annotations from the manually annotated test corpus were available for one [8]. The models themselves were not made available so could not be evaluated on our corpus. TABoLiSTM (BioBERT) evaluated on the test annotations (1856 entities) from [8] resulted in a recall of 0.81 which outperforms the recall of 0.74 obtained by the authors. However, the test annotations also include several entities that are not metabolites, such as chemical elements/ions (calcium, ${\mathrm{Fe}}^{2+}$, H(+), nitrogen, oxygen, potassium, sodium and sulfur) and enzymes (aminoacyl-tRNA synthetases, asparagine synthetase, lactaldehyde dehydrogenase and vacuolar proton ATPase); Excluding these gives a total of 1815 entities that are metabolites (including abbreviations) and applying TABoLiSTM resulted in a recall of 0.84 for these data. Excluding the 317 abbreviations of the test annotations resulted in the TABoLiSTM recall of 0.92 (evaluated on 1498 entities). This exemplifies two aspects, first, while the training data for TABoLiSTM does include some abbreviations (insofar the definition was found as a metabolite using the Auto-CORPus abbreviation file) it is less capable of recognising these in the absence of the definitions. This highlights further improvements can be made with better abbreviation detection algorithms. Second, the manual test annotations from [8] include several entities that are not metabolites. Our metabolomics test corpus contains over 19K metabolites, each of which are listed in HMDB [12] or found by rule-based methods, and the corpora (training and test) contribute both the sentences and annotations.

Our methods, compared to similar methods for chemical entity recognition [17,21,27] and metabolite NER [8,9], have the benefit of having been trained not only on abstracts and titles, but also on full-text paragraphs that are relevant for information retrieval from studies (results and discussion sections). The corpora these algorithms were trained and evaluated on are made available for future re-use and algorithm development, and to the best of our knowledge is the first of its kind for metabolomics. In addition, we developed a semi-automated annotation pipeline combining regular expression and rule-based approaches with dictionary searching, both for processing the training data for and as an alternative to DL methods, which too achieved outstanding performance (F1-score > 0.88) and also surpassed prior metabolite NER work.

### 3.1. Annotation Pipeline

Recent work that used NLP methods via dictionary searching for text mining in metabolomics highlighted the considerable user effort (1–4 h as per [10]) required to run these types of analyses. Our annotation pipeline consists of a dictionary of metabolite names and synonyms (from HMDB) which can easily be added to by including data from other databases such as ChemSpider [30] and ChEBI [31]. However, any entity added to the dictionary must undergo some form of quality control to ensure it is a metabolite. Here, we have limited ourselves to only including metabolites listed in HMDB that were classified as either ‘quantified’ or ‘detected’ and have not included those that are ‘expected’ or ‘predicted’.

The annotation pipeline also includes regular expression and rule-based methods used to recognise metabolites in text and these add additional advantages: they are time-efficient, easily customisable and explainable. While it requires minutes or even hours for a skilled expert to annotate paragraphs and entire full-text articles, the time to process one full article using these methods is less than 15 s on average. Once experts establish additional rules focusing on recognising naming conventions, then these can easily be added into the pipeline. Moreover, for each identified entity it is immediately obvious which rule(s) resulted in it being predicted as a metabolite entity, and therefore more interpretable than a DL model. For this reason, any false positives or false negatives can be analysed and new sets of rules established to correctly annotate the metabolites in text. This also allows for developing a user-feedback pipeline in which new rules can be shared to obtain better results. We have used this semi-automated annotation pipeline to prepare the training corpus for the DL methods, therefore any improvement in the rules for annotation may also improve DL models. Another application of the pipeline is to run it in parallel with the DL-based methods and combine the results as a post-processing step.

The main limitation of the annotation system is that the system assumes that the dictionary of metabolite names contains all known metabolites with irregular names. We used the HMDB as dictionary because it is the most comprehensive database of human metabolites thus far, however our annotation approach may still result in missing metabolites. One example is endoxifen, which is an active metabolite of the breast cancer drug tamoxifen. Although its synonym 4-hydroxy-*N*-desmethyltamoxifen can be recognised using regular expressions (for example because of the prefix, see Appendix A), the word endoxifen itself is not recognised by the pipeline because of its irregularity and not being included in our dictionary (it exists in HMDB as ‘expected’). Augmenting the dictionary with the metabolites listed as ‘expected’/‘predicted’ may appear to be a feature and obvious solution, however adding all of these terms would generate a dictionary 4 times larger than the current one and therefore drastically increase the run-time for the rule-based annotation. Also, >90% of the ‘expected’ metabolites are lipids [12], and these names are usually recognisable using regular expressions.

Overall, the annotation pipeline is dependent on its data and rules, it can therefore not detect metabolites that have been misspelled, nor will it flag certain terms as false positives if they appear in the dictionary as synonyms for a metabolite. For example, common words or words with multiple meanings (depending on the context) would be recognised by the system such as ‘result’ (synonym of ‘omeprazole’) and ‘retinal’ (both a vitamin-derivative and a medical term for ‘eye’). HMDB also includes chemical entities such as ‘ammonium acetate’ and ‘silica’, which are commonly mentioned in methods sections of metabolomics articles. As the HMDB includes more than 18,000 ‘detected’ and ‘quantified’ metabolites and their synonyms, it was not feasible to examine all entities as pre-processing in this study and only those with 5 or less characters were manually checked; therefore, further cleaning of the dictionary would likely improve the precision of the annotation pipeline. Our results indicate that the combination of regular expressions, dictionary searching and rules can indeed correctly capture a vast amount of metabolite names as also reported by Majumder et al. [10].

### 3.2. Metabolomics Corpus

To the best of our knowledge, our metabolomics corpus, which consists of the abstract, method, result and discussion sections of 183 manually annotated (test set) and 1034 semi-automatically annotated (training set) PMC articles, is the first available corpus that is designed for human metabolomics research. While some databases include some references to articles where metabolites are mentioned, the databases do not contain enough information (e.g., annotated texts) for NLP model development. An advantage of our metabolomics corpus, compared to common chemical NER corpora [15,16], is that it contains not only abstracts but also sentences from the method, result and discussion sections which is relevant for context-dependent NER. While abstracts indeed provide a rich set of context, the structure of sentences is often different from the main text and therefore training and employing contextual NLP models only on abstracts could potentially result in overlooking valuable information while mining the full text of articles. Methods sections are included since these can include information on targeted metabolites with different assays.

The limitation of the current metabolomics corpus is that it was designed only to label metabolites that appear in full in sentences; that is, if the recognition of a metabolite requires semantic interpretation then it would not be correctly detected. For example, if the string ‘1- and 3-methylhistidine’ is encountered by the pipeline, then only ‘3-methylhistidine’ is annotated, whereas the metabolite ‘1-methylhistidine’ is also in the string. To overcome this limitation of the corpus, it needs to be processed further using semantic analysis techniques such as parse trees [32,33].

We are currently working on addressing a limitation of the Auto-CORPus package [29] that we used to process the full-text. In investigating our results, we found that any abbreviations that cannot be mapped to full names are not annotated, this includes for example cases where Greek letters are abbreviated by letters of Latin alphabet (α as ‘a’). We also encountered instances where metabolites are abbreviated without the abbreviation (PC, PE, SM, and others) being defined, because the abbreviations are commonly used in the (lipidomics) community. These abbreviations were also not annotated in the (semi-automatically annotated) metabolomics corpus, but could easily be added as a set of rules to the algorithm. Lastly, superscripts and subscripts in the corpus are encoded differently from normal text by Auto-CORPus [29] and this causes negative impact on the annotation algorithms. We therefore anticipate periodically updating the metabolomics corpus based on user feedback and further research.

### 3.3. Deep Learning Models

Four LSTM-based models were evaluated on the same manually annotated test set as the annotation pipeline. ChemListem [21], which achieved an F1-score > 0.90 on the BioCreative V5 [16] challenge data, achieved a lower F1-score of 0.77 when evaluated on the metabolomics corpus test set. This is in contrast with our MetaboListem model that used the same neural network architecture as ChemListem, but was trained on the metabolomics corpus, achieving an F1-score of ∼0.89. The lower precision and recall rates of ChemListem, when applied to our corpus, are the results of higher false positives and false negatives, respectively. The false positives are expected due to the fact ChemListem was trained to recognise chemical entities and not metabolites specifically. The higher false negative rate is indicative of limited power for detecting metabolites—despite these also being chemical entities. The recall rates of ChemListem evaluated on the CEMP and metabolomics corpora are 0.89 [21] and 0.81 (here), respectively, which is indicative of the fact that metabolites appear less frequently in the BioCreative V5 dataset. Our MetaboListem model outperforms ChemListem on all three metrics for the metabolomics corpus, thus indicating that the BiLSTM network structure is generalisable to finer NER tasks and that training on relevant data further improves performance. This allows the algorithms to deal with changes in nomenclature and overcome the issue of having to define complex domain knowledge with limited sets of rules which was observed in NLP tasks for recognising genes and proteins previously [34]. Moreover, we have shown how each algorithm achieves different levels of performance on individual sections of a publication, therefore algorithms developed only on abstracts (as these are easily available) may not perform as well on full-text sections. Our algorithms have been trained on a variety of sections (including abstracts) and have shown good performance across all sections.

Our second model, TABoLiSTM, replaced the GloVe embedding [23] that was used in MetaboListem with BERT [24] and BioBERT [25]. The TABoLiSTM model with BioBERT achieved the highest F1-score (0.91) of all models evaluated and improves on all metrics compared to MetaboListem. This improvement is directly attributable to using contextualised token embeddings—enhancing the contextual sensitivity—and using smaller token sequences that are in between character- and word-level embeddings. Smaller token sequences are known to be more suitable for NER tasks where segmenting words is difficult [35], for example with metabolites.

While the annotation pipeline has the highest recall rate, we observed that for many instances with spelling errors in metabolite names that these were recognised by the MetaboListem and TABoLiSTM models but were missed by the rule-based annotation pipeline. This includes for example misspelled entities such as ‘slanine’ (typo of ‘alanine’), ‘acetoace acid’ (typo of ‘acetoacetic acid’) and ‘palmi acid’ (typo of ‘palmitic acid’) indicating that the DL models are capable to generalise and identify unseen metabolite names, including the misspelled ones, by recognising the token structures and the contexts. In light of earlier discussion on the annotation pipeline not recognising endoxifen, the DL models do correctly recognise endoxifen as metabolite which is likely because the token structure of endoxifen is similar to that of tamoxifen in addition to the similar context embedding of metabolites.

MetaboListem outperforms the precision of the annotation pipeline by 0.7% which may be explained by the ambiguous synonyms falsely identified by the annotation pipeline. For the TABoLiSTM model (BioBERT embedding achieving higher precision by 4% compared to the annotation pipeline) this may be to learning contexts rather than learning the rules and regular structures designated to the annotation pipeline. The algorithms were (trained and) evaluated on the full text output from Auto-CORPus [29], however Auto-CORPus also provides separate JSON output files for table data and abbreviations and these files mostly contain single terms without context. Empirically, we found that although the DL models are context sensitive by construction (BiLSTM network and BioBERT embedding) they detect entities in tables and abbreviation lists with high accuracy comparable to the full text results.

Overall, the semi-automatic annotation pipeline is ideally used in parallel with the DL methods as combining these can to provide wider coverage of features that are not yet learned by the DL algorithms due to limited training examples. Likewise, this is also relevant for addressing some limitations of the DL methods identified from their false positives. We found that while most bacteria are not recognised, a very small number of bacterial species are picked up as ‘metabolites’ by the DL methods. These terms all include tokens that can also be found in metabolites such as ‘lact’ (*Lactobacillus*, *Lactococcus*), ‘butyr’ (*Butyricimonas*, *Pseudobutyrivibrio*, *Butyrivibrio*), ‘chloro’ (*Chloroflexi*), ‘succini’ (*Succinivibrio*), ‘sulfo’ (*Desulfovibrio*), and ‘acid’ and ‘amino’ (*Acidaminobacter*). As part of a wider effort to provide NLP tools to the omics community, we are developing an NER algorithm for microbiome studies which can also be used to filter out false positives from our DL models when complete. Other metabolites are identified (by both the DL and rule-based methods) that are in fact part of an enzyme or pathway name. In our DL models we have not included negative training examples of terms in these contexts, however more elaborate rules can also be developed to filter these false positives out in a post-processing step. For example, we observed in the training set that the words ‘process’ and ‘success’ are falsely recognised as metabolites in a certain context. Based on the dictionary list, no known metabolite name/synonym ends with ‘ss’, so therefore all terms that match the regular pattern ‘ss$’ could be excluded. This type of rule-based post-processing was not used for our models because of the risk of introducing bias in model evaluation, where an n-gram of ‘success’ is found in metabolite names such as succinic acid/succinate.

In terms of usability of the DL algorithms, the model size is an important factor. The TABoLiSTM models require 827 MB of storage, whereas ChemListem and MetaboListem only need 26 MB. Therefore, TABoLiSTM may be too heavy to be leveraged on web services and would ideally be used on local systems. Therefore, employing the lighter MetaboListem model in an online environment is more feasible in future, regardless of the cost of 1.8% in the F1-score relative to the TABoLiSTM (BioBERT) model (which outperforms MetaboListem on all evaluation metrics). On average, a full-text article can be annotated by MetaboListem in under 10s, whereas for TABoLiSTM this is close to 40 seconds, hence users may find MetaboListem easier to use. The annotation pipeline and TABoLiSTM (BioBERT) could be used together to increase the confidence of automated annotations (where both methods are in agreement) as they are based on separate methodologies. However, the recall rates are virtually identical and the precision of TABoLISTM is over 4% higher compared to the pipeline, hence it is doubtful this will increase the overall performance of a ‘hybrid’ algorithm. Moreover, the annotation pipeline requires users using the most up-to-date dictionary at all times to achieve the best results whereas our DL methods have already shown they are capable of generalisation (e.g., correctly recognising metabolites with spelling errors). Hence, the contribution of TABoLiSTM (and MetaboListem) here is to provide an alternative to the reliance upon dictionaries of other NER algorithms [10]. However, as we have shown by using different abbreviation detection algorithms in Auto-CORPus [29] this can be used to increase the number of annotations, therefore an open line of investigation is to evaluate whether the confidence of users in the accuracy of automatic (opposed to manual) annotations is increased if the annotations are identified with multiple (different) methods.

### 3.4. Biological Interpretation

We have exemplified how the algorithms can be used to speed up literature review by providing a summary of all identified metabolites in the full text. The most common metabolites reported appear to be those that can be measured using virtually all major assays. Glucose is reported most frequently for 4 of the traits, and as one of the top 5 metabolites for the other traits. For cancer studies, the high co-occurence of glucose and lactate is indicative of the Warburg effect [36] where glucose is converted to lactate for cancer cell energy metabolism. However, the remaining top reported metabolites are mostly amino acids, which is also observed for other traits. At the same time, some trait-specific metabolites are found such as bilirubin and cholesterol for liver diseases, cholesterol and triglycerides for metabolic syndrome, cotinine and nicotine for smoking, and tryptophan, kynurenine and serotonin for neurological/psychiatric/brain diseases. The latter being reflective of the serotonin hypothesis [37] that has resulted in antidepressant drugs such as kynurenine uptake inhibitors. The proximities between nodes in the network graphs are known to express community structures [38], hence it is not surprising to find so many amino acids and other common metabolites near the centre of the graphs.

## 4. Materials and Methods

### 4.1. Dataset

The data used in the study consists of Open Access metabolomics publications from PMC (*n* = 1217). The metabolomics corpus is constituted of human-related articles in 18 categories; 10 are by journals (see Table 1) and 8 of which are selected by traits (see Table 2). The complete search terms are provided in Appendix A.

The metabolomics articles were stored in HyperText Markup Language (HTML) format, which were then processed by the Auto-CORPus package [29], and standardised into machine-readable JavaScript Object Notation (JSON) documents.

Auto-CORPus outputs three JSON files based on the input of an HTML file of an article: ‘maintext’, ‘table’ and ‘abbreviation’. In the maintext file, the textual content in the HTML is split into subsections, and each subsections have five attributes: ‘section_heading’, ‘subsection_heading’, ‘body’, ‘IAO_term’ and ‘IAO_ID’, where IAO stands for Information Artifact Ontology [39]. Here we focused on four section types based on standardised IAO terms, namely the textual abstract (‘A’),methods (‘M’), results (‘R’) and discussion (‘D’) sections. We did not use the introduction sections as these do not typically contain information pertaining to the study itself or (how) new results (were acquired).

### 4.2. Rule-Based Annotation Pipeline

The metabolomics corpus was annotated using a semi-automated pipeline to deal with the large volume of texts and prepare it for supervised learning. The pipeline takes the Auto-CORPus-processed articles as JSON input, and outputs the location (character number) of metabolites in each text (an example can be found in Figure 1 in the Results section). The annotation pipeline is comprised of three steps: pre-processing, identifying metabolites and locating recognised entities.

#### 4.2.1. Pre-Processing

(Sub)sections are split into sentences using sentence tokenisation with the BioBERT [25] model (version biobert-base-cased-v1.2) using the Huggingface Transformer package [40]. Each sentence is uniquely identifiable by two attributes: the PMC identifier (PMCID) of the article that the sentence belongs to, and a sentence identifier. A sentence identifier is a string that contains a letter and five digits (Figure 5) where the letter indicates the relevant sections types (textual abstract (A), methods (M), results (R) and discussion (D)). These section types are determined by the IAO_term key in the Auto-CORPus full-text JSON file.

#### 4.2.2. Metabolite Identification

The next step is to identify any metabolite mentioned in each tokenised sentence using both dictionary searching and regular expression matching. The dictionary used here was based on the list of metabolites (and their synonyms) that are classified in HMDB [12] as ‘quantified’ or ‘detected’. A list of all metabolites and synonyms was first downloaded on 25 June 2020, and subsequently refreshed on 16 August 2021, after which it was filtered for the ‘quantified’ or ‘detected’ metabolites. Short abbreviations (up to 5 characters) that are not exclusive for metabolites were removed (e.g., ‘PC’ can refer to the metabolite ‘phosphatidylcholine’, but is also a common abbreviation for ‘principal component’ and ‘pancreatic cancer’), and it was then further cleaned of other non-specific words such as ‘result’ (a synonym of ‘omeprazole’).

To complement dictionary searching, the regular expressions are designed to provide partial matches for the metabolite entities (Table A1). It is expected that more metabolites can be captured in this manner compared to complete matching. Entities with a partial match are then expanded to include the entire word and included a full metabolite name or part of a metabolite name; the latter scenario is treated further in post-processing. To craft the regular expressions, the HMDB dictionary was used as a reference. From the dictionary, the entities with one or more spaces and the ones with symbols like hyphen (‘-’) and colon (‘:’) were selected to form a ‘reference set’. The reference set meant to include metabolite names that are less trivial (and thereby more regular). Trivial terms such as singletons are usually more difficult to be captured with regular expressions without drastically increasing the number of false positives by including common words. The regular expression approach is used to complement the HMDB dictionary, therefore trivial terms are omitted from the reference set as they are assumed to be listed in HMDB already. This was used to create a set of regular expressions such that most, if not all, of the reference set can be (at least partially) matched by at least one regular pattern in the set, while avoiding matching unwanted terms.

The regular expressions were created in a recursive manner: while not all terms in the reference set are matched and there remain some observable patterns in the remaining terms, either a new expression was created or an existing expression augmented in order to match the remaining regular terms. Examples of regular expressions and the terms that they are expected to match are shown in Appendix A.

#### 4.2.3. Post-Processing

Post-processing of the matches was utilised to integrate the result of dictionary searching and regular expression matching, including those from partial matching. This step is comprised of three components: combining adjacent words, balancing the number of brackets, and merging overlapped entities.

For each metabolite entity that has been recognised its adjacent words are evaluated. The term before the entity is prepended to the annotation only if certain rules are satisfied: the preceding word ends with a hyphen (‘-’), ends with a number, the recognised entity itself starts with a comma and number (e.g., ‘,3’) or the current word starts with a hyphen. If the preceding word is a stop word, or if none of the rules can be met, then the procedure is complete an no term is prepended to the entity. The rules for the next word are the same as described above, except now the entity becomes the ‘preceding’ word in the above scenario and one more rule is evaluated (whether the next word is one of either ‘acid’, ‘isomer’, ‘ester’ or ‘ether’). This process is recursive and is done until no adjacent words satisfy any of the rules.

For all entities the number of brackets is evaluated to ensure all parentheses, square brackets and curly brackets have a starting and ending bracket in the entity. If an opening or closing bracket is missing then the entity is enclosed by the corresponding missing bracket.

All entities that overlap or separated only by a (white)space character are merged into a single entity. The entire post-processing step is executed recursively, i.e., until there is no more change in the list of identified metabolite entities. Once this has finished, two files are created that are used to train the DL models. The first file contains all sentences that have at least one recognised metabolite, along with their sentence identifiers (see Figure 5) and the PMCIDs of the articles the sentences originate from. The second file is in table with 5 fields: PMCID, sentence identifier, position of the start character of the entity, position of the end character of the entity, and the entity itself. These two files are exemplified in Figure 1B,C.

### 4.3. Training, Validation and Test Set Generation and Evaluation

The 1218 metabolomics publications were split at random into a training, validation and test set a 75:10:15 ratio, corresponding to 913 training publications, 122 publications for validation and 183 publications for the test set. The publications in the training and validation set were processed using the rule-based annotation pipeline described above, creating a semi-automatically annotated corpus. The TrainingSet.txt and TrainingSetAnnot.tsv (see https://github.com/omicsNLP/MetaboliteNER, accessed on 15 March 2022) files contain both the training and validation partitions as these are split within the training process by the algorithms. The publications in the test set, this included entities in the full text as well as in tables, were manually annotated to avoid bias and to provide a ground truth to evaluate the algorithms on.

Although metabolite abbreviations and entities in tables were not deliberately annotated in the training set with the annotation pipeline, they were manually annotated in the test set for completeness; these types of metabolite names were considered separately while employing models and evaluating performances (detailed in Post-processing). The performance of algorithms was assessed by evaluating the number of true positives (*TP*), false positives (*FP*) and false negatives (*FN*) and calculating the precision ($\frac{TP}{TP+FP}$), recall ($\frac{TP}{TP+FN}$), F1-score ($\frac{2\times precision\times recall}{precision+recall}$) and F*-score [41] ($\frac{precision\times recall}{precision+recall-precision\times recall}$) on the test set.

### 4.4. Metabolite NER Using LSTM

A BiLSTM model architecture that was used previously on chemical NER task achieving F1-scores > 0.9 [21] was used here for metabolite NER. We first evaluated the original ChemListem model on the corpus, we then developed two new methods (MetaboListem, TABoLiSTM) based on the prior methodology to improve the performance and described below. Our new models were trained in 50 epochs, and the epoch that achieved the highest F1-score on the validation set was used for evaluation of the test set. All F1-scores reported here are for the test set only.

#### 4.4.1. Pre-Processing

Following prior work [21], a tokeniser designed for chemical text-mining (OSCAR4) [42] was employed to tokenise words in the metabolomics corpus for MetaboListem. For the TABoLiSTM model, we used BioBERT as tokeniser [40] because TABoLiSTM uses the BioBERT token embeddings.

For both word tokenisation systems, the BIOES tag scheme was employed to label token sequences. BIOES tag scheme is a standard sequence labeling technique that classifies tokens into five categories: ‘B’ labels the tokens at the beginning of entities, ‘I’ labels internal tokens of entities, ‘E’ labels the ending tokens of entities, ‘S’ labels singletons (i.e., tokens that are a complete entity on their own), and ‘O’ the tokens that do not belong to an entity.

ChemListem used a pre-classifier random forest to predict the probability of a token being (part of) a chemical entity (‘B’, ‘I’, ‘E’ or ‘S’) based on a set of ChEBI-derived [31] chemicals and chemical elements. For MetaboListem and TABoLiSTM we implemented the same pre-classifier approach using the set of metabolites from HMDB [12] rather than chemicals from ChEBI, where tokens were segmented using OSCAR4 and BioBERT for MetaboListem and TABoLiSTM, respectively. The pre-classifier produces name-internal features that were used in the model training.

#### 4.4.2. ChemListem and MetaboListem

For ChemListem [21], name-internal features for each token were computed in the pre-processing stage using the pre-classifier and treated as one of the two inputs to the model (Figure 6). The features go through a convolutional layer with width of 3 and are concatenated with the numeric representation of words (i.e., a unique number assigned to each word) in the GloVe [23] vector space. Each token now has 256 + 300 = 556 dimensions, where 256 is the output neuron number of the convolutional layer, and 300 the GloVe vector representation. The concatenated tensor is then fed into the BiLSTM layer, before outputting a 5-dimensional result (number of tags used in BIOES tagging) via a TimeDistributed Dense neuron.

Our MetaboListem algorithm is based on the same neural network architecture (and GloVe embedding) as ChemListem, hence we named the algorithm in the same manner. It differs from ChemListem only in the training data (metabolomics corpus opposed to BioCreative V5 [16]), pre-classifier data (HMDB instead of ChEBI) and TensorFlow implementation (version 2.0 versus 1.3).

#### 4.4.3. TABoLiSTM (Transformer-Affixed Bidirectional LSTM)

Our TABoLiSTM model differs from MetaboListem mainly in the use of pre-trained BERT [24] (bert-base-cased) and BioBERT [25] (biobert-base-cased-v1.2) embedding. The name-internal features and the numeric presentation of words were all re-computed using (Bio)BERT tokens instead of words and the last hidden layer of the (Bio)BERT model replaces the GloVeEmbeddingLayer in Figure 6. The (Bio)BERT layer provides contextualised word embeddings, and the embeddings were concatenated with the convolution output of name-internal features as previously, except for larger tensors of 256 + 768 = 1214 dimensions. A SpatialDropout1D layer with dropout rate 0.25 is applied on top of the (Bio)BERT layer to prevent overfitting and improve model performance by embedding the dropout [43].

#### 4.4.4. Post-Processing

Post-processing of our DL models involves the same procedure as that of the annotation pipeline (Section 4.2.3) to fix pathological terms such as incomplete entities: to check the neighbouring entities, to fix unbalanced parentheses if possible, and to merge overlapped positions. At this stage further rule-based approaches can be employed to further improve models, for MetaboListem and TABoLiSTM these were not used, but rather represent potential further improvements that can be made to the post-processing.

The DL algorithms were trained on sentences with full metabolite names only, and were not trained on abbreviations or data in tables. In the post-processing step, the algorithms were applied to the Auto-CORPus JSON output of the list of abbreviations (abbreviations and definitions) and any recognised entity (definition) was then linked to its abbreviation and all occurrences in the full text marked as a recognised entity. This was only done when the entire definition was recognised as metabolite entity, i.e., the rules to fix incomplete entities are not applied here. A direct approach to replace all abbreviations in the text with their definitions was investigated but discarded since the current implementation of the abbreviation recognition does not detect all types of abbreviations. The same approach was undertaken on the Auto-CORPus JSON output of all tabular data, where each cell was evaluated by the DL models as a sentence. Any recognised entity was annotated and evaluated against the manually annotated entities in the test set.

### 4.5. Metabolite Co-Occurence Network

TABoLiSTM (BioBERT) was applied to all articles in the corpus related to cancer and smoking, and only on the abstract, result (including those in tables) and discussion sections. For each pair of (unique) metabolites mentioned in an article the edge is weighted, likewise the node weight is increased for each (unique) metabolite found. Using these data an adjacency list was created and imported into Gephi [44]. The graph layout was optimised using the ForceAtlas2 [45] algorithm and visualised without edges for the sake of visibility. Nodes placed closer together are more likely to be mentioned in the same articles, and the size of the text is directly proportional to the number of times the metabolite is mentioned across all articles of the same trait. For the cancer graph only the metabolites that appear in three or more articles are used for the graph construction because of the large number of unique metabolites reported (see Table 2). For the smoking graph all metabolites (including those only found by a single study) are displayed. The node names are coloured based on the article that reports the metabolite most (if such article is not unique, the one with lower PMCID number is selected).

## 5. Conclusions

In summary, this work contributed the first annotated metabolomics corpus of full-text articles to facilitate future NER development for metabolomics research. This corpus is split into two parts: a training set with 49,007 sentences from different sections (abstract, methods, results, discussion) that contains 105,335 annotated metabolite entities, and a (gold-standard) test set with 8950 sentences and 19,138 manually annotated metabolites. This will allow future work on metabolite NER to be trained and/or evaluated on the same data to allow direct comparison of performance between different algorithms. Our four novel NER algorithms, including a rule-based annotation algorithm and three DL NER systems (each with different word embedding layers), are implemented in Python. These algorithms have all shown state-of-the-art performance (F1-score > 0.88) for metabolite NER tasks with the best model (TABoLiSTM with BioBERT word embeddings) achieving an F1-score of 0.91, thereby facilitating future effort in text-mining metabolomics literature. Current on-going work by us focuses on developing new corpora and algorithms for enzyme, pathway and microbiota NER, each of which will allow more effective data extraction and these are intended to further improve the performance of TABoLiSTM by filtering out any false positives or better indicate the context of the metabolite (part of an enzyme or pathway name opposed to a metabolite on its own). Fine-tuning a bidirectional transformer model such as BioBERT for metabolite NER is another potential avenue worth exploring using the metabolomics corpus. All code and data is made available to allow other researchers to use or build upon our work to enable researcher to more effectively perform literature review for metabolomics research.

## Figures and Tables

**Figure 1 metabolites-12-00276-f001:**
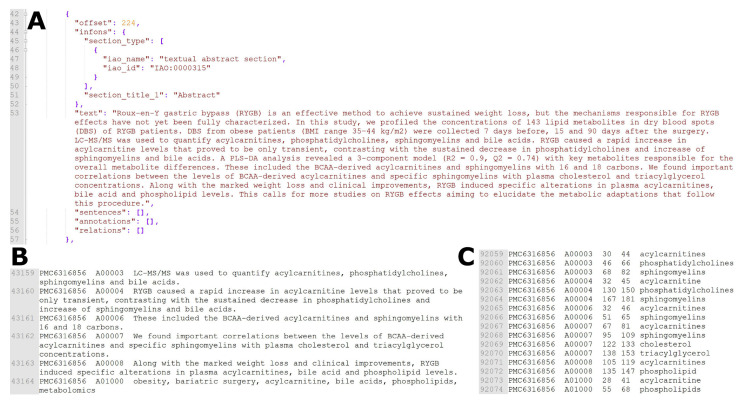
Data processing and corpus generation. (**A**) The Auto-CORPus output (BioC JSON format) for the abstract of PMC6116856. (**B**) Extracted sentences containing annotations in the abstract (part of the metabolomics corpus, training set). (**C**) Rule-based annotations for the abstract sentences (part of the metabolomics corpus, training set).

**Figure 2 metabolites-12-00276-f002:**
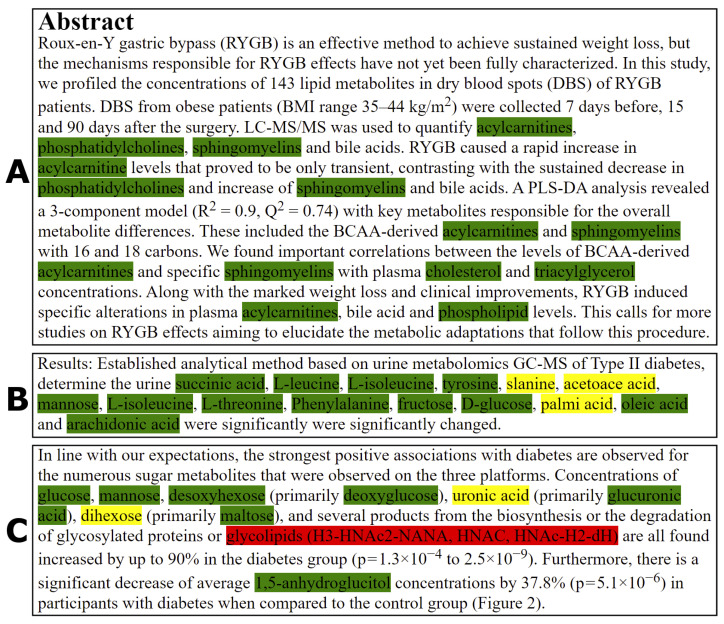
Examples of TABoLiSTM and rule-based annotation results visualised in annotated HTML files. Entities marked in green are recognised by both TABoLiSTM and the rule-based annotation pipeline, yellow entities are only recognised by TABoLiSTM, and red entities only by the rule-based annotation. (**A**) Excerpted from PMC6116856. (**B**) Excerpted from PMC4969426 and exemplifies typos. (**C**) Excerpted from PMC2978704.

**Figure 3 metabolites-12-00276-f003:**
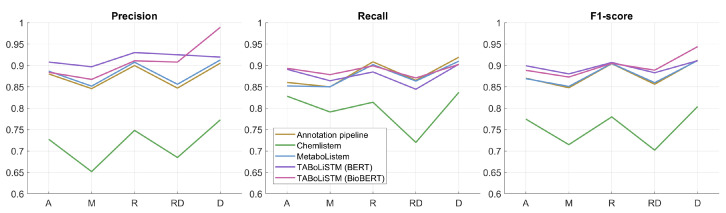
Model performance per publication section. The model performance (precision, recall and F1-score) are given for 5 sections (A, abstract; M, materials/methods; R, results; RD, results and discussion (as single section); D, discussion) for the 5 models.

**Figure 4 metabolites-12-00276-f004:**
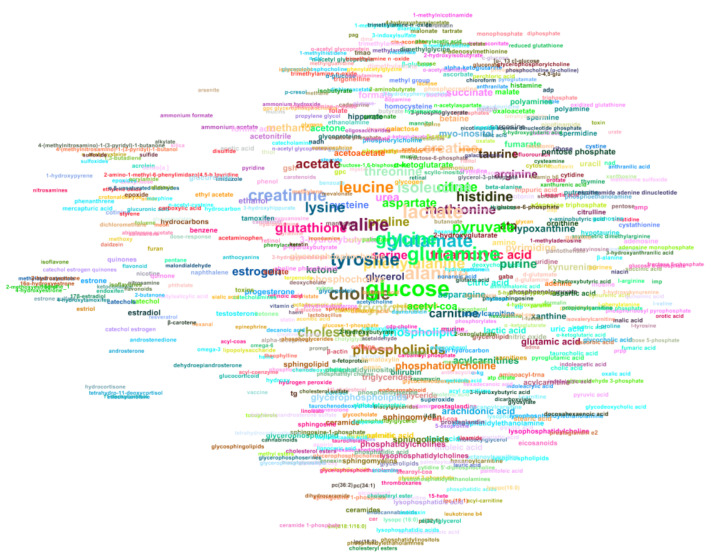
Graphical representation of metabolites mentioned in the abstract, results and discussion sections of 492 cancer-related metabolomics articles. Each node represents a metabolite and is displayed as the (lower-cased) metabolite name if it appeared in 3 or more articles. The closer metabolites are in the graph, they are more likely to co-occur in the same article. The text is sized proportionally to the number of articles that report the metabolite.

**Figure 5 metabolites-12-00276-f005:**
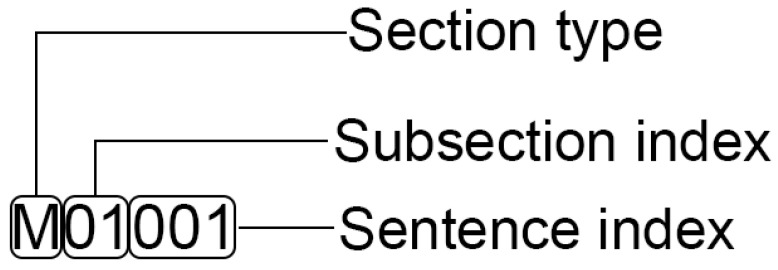
An example of sentence ID. The ID refers to the second sentence in the second subsection of the method section. Indexes start with 0.

**Figure 6 metabolites-12-00276-f006:**
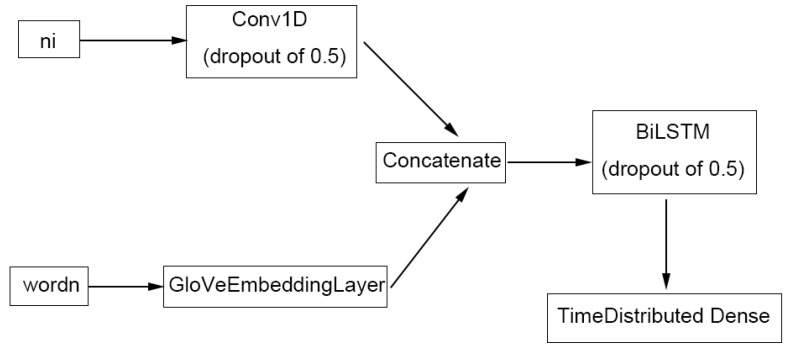
The architecture of the ChemListem network. The input tensors ‘wordn’ and ‘ni’ correspond to the numerical representations of word (token) sequences and the name-internal features of word (token) sequence determined by the pre-classifier, respectively.

**Table 1 metabolites-12-00276-t001:** Summary of number (#) of unique metabolites mentioned per journal. Each metabolite is only counted once per article.

Journal	# of Metabolites	# of Articles	Average # of (Unique) Metabolites per Article
*Analytical and Bioanalytical Chemistry*	716	34	21.1
*Analytical Chemistry*	863	48	18.0
*Journal of Chromatography A*	220	9	24.4
*Journal of Proteome Research*	694	24	28.9
*Metabolites*	1279	50	25.6
*Metabolomics*	1267	43	29.5
*PLOS One*	5400	175	30.9
*PNAS USA*	429	14	30.6
*Scientific Reports*	4278	126	34.0

**Table 2 metabolites-12-00276-t002:** Summary of the number (#) of unique metabolites annotated per trait. Each metabolite is only counted once per article.

Trait	# of Metabolites	# of Articles	Average # of (Unique) Metabolites per Article
cancer	12,662	492	25.7
gastrointestinal	876	37	23.7
liver disease	1959	121	16.2
metabolic syndrome	7081	286	24.8
neurodegenerative, psychiatric, and brain illnesses	2546	113	22.5
respiratory diseases	398	37	10.8
sepsis	552	22	25.1
smoking	1360	124	11.0

**Table 3 metabolites-12-00276-t003:** Summary of model performance. Models are evaluated in terms of F1-score, precision and recall rate, and F*-score using the metabolomics corpus test set (183 manually annotated full-text articles). The best results for each metric is highlighted in bold.

Model	Training Data	Embedding	F1-Score	Precision	Recall	F*-Score	Model Size
Annotation pipeline			0.8893	0.8850	**0.8936**	0.8006	
ChemListem [21]	CEMP BioCreative V.5	GloVe	0.7669	0.7301	0.8075	0.6219	26 MB
MetaboListem	metabolomics corpus	GloVe	0.8900	0.8923	0.8877	0.8018	26 MB
TABoLiSTM	metabolomics corpus	BERT	0.9004	0.9187	0.8829	0.8189	827 MB
TABoLiSTM	metabolomics corpus	BioBERT	**0.9089**	**0.9255**	0.8928	**0.8329**	827 MB

**Table 4 metabolites-12-00276-t004:** Top ten metabolites reported for each trait in the metabolomics corpus. The number of articles that report the metabolite are given including the percentage of the total. The total number of articles in each category can be found in Table 2.

Cancer		Gastrointestinal		Liver Disease		Metabolic Syndrome	
glucose	142 (29%)	glucose	12 (32%)	glucose	29 (24%)	glucose	178 (62%)
glutamine	121 (25%)	tyrosine	12 (32%)	cholesterol	25 (21%)	cholesterol	108 (38%)
lactate	117 (24%)	lactate	11 (30%)	creatinine	21 (17%)	valine	83 (29%)
alanine	111 (23%)	acetate	10 (27%)	tyrosine	20 (17%)	leucine	73 (26%)
glutamate	109 (22%)	phenylalanine	9 (24%)	glycine	18 (15%)	tyrosine	71 (25%)
tyrosine	105 (21%)	leucine	9 (24%)	phenylalanine	17 (14%)	triglycerides	70 (24%)
glycine	105 (21%)	tryptophan	9 (24%)	lactate	16 (13%)	alanine	67 (23%)
valine	104 (21%)	alanine	8 (22%)	bilirubin	16 (13%)	isoleucine	65 (23%)
choline	103 (21%)	arginine	8 (22%)	pyruvate	15 (12%)	phenylalanine	62 (22%)
creatinine	95 (19%)	citrate	8 (22%)	valine	15 (12%)	glycine	59 (21%)
**Neurodegenerative, Psychiatric, and Brain Illnesses**		**Respiratory**		**Sepsis**		**Smoking**	
tryptophan	31 (27%)	lactate	8 (22%)	phenylalanine	11 (50%)	creatinine	24 (19%)
glucose	26 (23%)	glycine	6 (16%)	glucose	9 (41%)	cotinine	15 (12%)
glycine	25 (22%)	leucine	6 (16%)	lactate	9 (41%)	nicotine	14 (11%)
glutamate	22 (19%)	pyruvate	6 (16%)	arginine	8 (36%)	glucose	14 (11%)
kynurenine	22 (19%)	glucose	6 (16%)	urea	6 (27%)	leucine	11 (9%)
phenylalanine	21 (19%)	lysine	5 (14%)	nitric oxide	6 (27%)	isoleucine	11 (9%)
serotonin	18 (16%)	creatine	5 (14%)	methionine	6 (27%)	valine	11 (9%)
tyrosine	18 (16%)	isoleucine	5 (14%)	pyruvate	6 (27%)	cholesterol	11 (9%)
creatinine	18 (16%)	glutamine	5 (14%)	creatinine	6 (27%)	lactate	10 (8%)
lactate	17 (15%)	acetate	4 (11%)	ATP	5 (23%)	glutamate	9 (7%)

## Data Availability

All data generated, including the annotated metabolomics corpus (manually annotated test set (GoldStandard.txt and GoldStandardAnnot.tsv) and semi-automatically annotated training and validation sets (TrainingSet.txt and TrainingSetAnnot.tsv)), the MetaboListem model and code for post-processing, is available via https://github.com/omicsNLP/MetaboliteNER (committed 24 February 2022), the BERT-embedded version of TABoLiSTM is available via https://doi.org/10.5281/zenodo.6340002 (created 9 March 2022).

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
