# Peer review of "MetaboListem and TABoLiSTM: Two Deep Learning Algorithms for Metabolite Named Entity Recognition"

_metabolites, 2022, doi:10.3390/metabo12040276_

Round 1

Reviewer 1 Report

The article is within the scope of the journal, and deals with an interesting topic.

It is well written and easy to read.

The results presented are original and represent an advance in the area of knowledge.

However, some aspects should be improved:
a) The state of the art should be extended in the introduction
b) The conclusions should be expanded essentially describing the scientific contribution done. Likewise, a set of lines of future work should be included.

Reviewer 2 Report

The manuscript entitled “MetaboListem and TABoLiSTM: Two Deep Learning Algorithms for Metabolite Named Entity Recognition” established the annotated metabolomics corpus based on the full-text paragraphs instead of the only abstracts section compared with the previous work. In addition, from this corpus, the authors trained two deep learning algorithms to perform metabolite NER. This work is interesting considering the wide applicability of metabolomics in various fields and presents a novel exploration in the text-mining of metabolomics literature. This paper can be improved if the following points are addressed.

  1. In line 232, what is the experimental design reason for using the method section to supplement the metabolomics corpus? This part typically does not contain metabolite information.
  2. It is suggested to use more standard three-line tables, clear figures (Figure 3), and more concise elaboration which would make the reader easily understand.
  3. Additionally, the manuscript requires moderate editing for grammatical mistakes.

Other minor issues are listed below:

Page 1, line 3, “…more efficient literature review.” should be changed to “…more efficient literature reviews.”

Page 2, line 82, “…as well as not constrain…” should be changed to “…as well as not constraining…”

Page 7, line 165, “…knowledge are the first of…” should be changed to “…is the first of…”

Page 7, line 173, “…these type of analyses.” should be changed to “…these types of analyses.”

Page 8, line 183, “…an skilled expert…” should be changed to “…a skilled expert…”

Page 8, line 234, “…the structure of sentences are often different…” should be changed to “… the structure of sentences is often different…”

Page 9, line 254, “…negative impact to…” should be changed to “…a negative impact on…”

Page 9, line 262, The sentence could be changed to “The lower precision and recall rates of ChemListem, when applied to our corpus, are the results of higher false positives and false negatives, respectively.” which is better for understanding.

Page 10, line 327, “…in certain context.” should be changed to “…in a certain context.”

Page 12, line 401, “…here was is based on…” should be changed to “…here was based on…”

Page 14, line 501, “…was computed in…” should be changed to “…were computed in…”

Reviewer 3 Report

The authors call for the introduction of two new deep-learning based
methods for the recognition of Named Entity Recognition (NER) methods,
together with a new corpus of domain knowledge, which has been used to
generated the models. The starting point for the proposal is a
Bidirectional Long Short-Term Memory (BiLSTM) architecture, enriched
with different transfer learning techniques: the first uses GloVe word
embeddings, while the second one exploits BERT and BioBERT resources.

The INTRODUCTION adequately contextualises and justifies the interest
of the proposal, through a brief presentation that also allows the
identification of the main issues with using these methods for
metabolite NER. However, an explicit description of the actual
contribution of the work on these issues is lacking. A short road-map
is clearly missing at the end of the section, which briefly introduces
the structure of the document and its contents.

The Section RESULTS should start with a brief description of the
contents of its subsections (three in total), before introducing the
latter. Impossible to guess simply from the title (Results) what it is
intended to convey. Therefore, here too a brief road-map is
necessary. The description of the metabolomic corpus (Subsection 2.1)
seems sufficient, although a comparison with the characteristics of
other similar corpora would serve to justify its use against those
alternatives, if they exist. The Subsection 2.2 makes it clear that
the paper does not introduce new deep-learning techniques, but simply
generates models adapted to the knowledge domain under consideration,
something that should be made explicitly clear in the ABSTRACT and in
the INTRODUCTION. In this sense, the expression (first phrase in
Section 3) "We have described three new DL models, using different
word embedding layers, ...", is the really appropriate one. 

The Section DISCUSSION includes conclusions that need to be
better explained. Thus, for example, the authors state that their proposal

"... surpass by a considerable margin previous methods (using CRFs)
for metabolite NER ..."

something that can only be asserted if the tests have been performed
under identical conditions. How can that be so, if the corpus
considered in this work is newly created ? While this aspect seems to
be clarified later (Subsection 3.3), the authors should have made this
clear from the outset.

It makes no sense at all to introduce the CONCLUSIONS Section before
the MATERIALS AND METHODS one, unless the latter is included as an
Appendix to the paper. I personally believe that MATERIALS AND METHODS
should be introduced before Section 2 (RESULTS).

In general, the texts at the foot of figures and tables are
excessively long. If an explanation of tables or figures is really
necessary, it should be included directly in the text.

On the other hand, while the results look promising, it would be
useful to compare them with some kind of gold-standard that can serve
as a reference. In the absence of such references, the authors should
reflect this in the text.

Round 2

Reviewer 2 Report

The authors have adequately solved all my questions.